# Suppression of interdiffusion-induced voiding in oxidation of copper nanowires with twin-modified surface

Chun-Lung Huang[1], Wei-Lun Weng[1], Chien-Neng Liao[1] & K.N. Tu[2]

Cavitation and hollow structures can be introduced in nanomaterials via the Kirkendall effect in an alloying or reaction system. By introducing dense nanoscale twins into copper nanowires (CuNWs), we change the surface structure and prohibit void formation in oxidation of the nanowires. The nanotwinned CuNW exhibits faceted surfaces of very few atomic steps as well as a very low vacancy generation rate at copper/oxide interfaces. Together they lower the oxidation rate and eliminate void formation at the copper/oxide interface. We propose that the slow reaction rate together with the highly effective vacancy absorption at interfaces leads to a lattice shift in the oxidation reaction. Our findings suggest that the nanoscale Kirkendall effect can be manipulated by controlling the internal and surface crystal defects of nanomaterials.

[1] Department of Materials Science and Engineering, National Tsing Hua University, 101 Section 2 Kuang-Fu Road, Hsinchu 30013, Taiwan. [2] Department of Materials Science and Engineering, University of California at Los Angeles, Room 3121D, Engineering V Building, Los Angeles, CA 90095, USA. Correspondence and requests for materials should be addressed to C.-N.L. (email: cnliao@mx.nthu.edu.tw)

The classic Kirkendall effect describes the interdiffusion of two components with different diffusion rates in a diffusion couple, which may cause marker motion, cavitation, or even internal stress[1]. In general, a vacancy flux will be generated in the opposite direction of the faster-moving component to balance the net atomic flux. If the vacancies can be completely absorbed by the crystal lattice, for example, by dislocation climb, an equilibrium vacancy concentration is maintained everywhere. Lattice shift (Kirkendall shift) occurs, which is manifested by marker motion, rather than the formation of porosity (Kirkendall void). Recently, hollow oxide or sulfide nanocrystals have been synthesized by the nanoscale Kirkendall effect[2]. A variety of hollow or core-shell nanostructures fabricated through Kirkendall voiding find many appealing applications in fuel cell, drug delivery, catalyst, biosensor, and plasmonics[3–7]. The major difference between the Kirkendall effect in bulk materials and nanoscale materials is due to the effect of lattice shift. In bulk materials, lattice shift can lead to vacancy equilibrium everywhere in the sample. In nanoscale materials, lattice shift tends to be negligible and the vacant lattice sites condense to form voids. Voids are undesirable for many nanoelectronic devices and nanoelectromechanical systems. For example, copper interconnects in very-large-scale-integration devices require a stable and void-free structure for performance and reliability considerations. Therefore, how to prevent Kirkendall void formation is of interest in nanotechnology. Understanding the nanoscale voiding kinetics becomes essential for controlling the void-free microstructure and the physical/chemical properties of nanomaterials.

Grain boundaries (GBs) are intrinsic crystal defects[8,9]. They are not only fast diffusion paths but also efficient sources and sinks of vacancies. A twin boundary is a special type of GB that preserves specific crystallographic orientation between twinned crystals. There are two types of twin boundaries in face-centered-cubic crystals: Σ3{111} coherent twin boundary (CTB) and Σ3{112} incoherent twin boundary (ITB)[10]. The CTB is a coincident-site-lattice boundary that has much lower energy than the ITB and the conventional high-angle GBs[11]. The influences of CTBs on dislocation glide, electron transport and atomic diffusion are different from those of ITB and high-angle GBs. Recent experimental and modeling work towards understanding the

effects of nanotwinned structure on mechanical and physical properties of metallic materials can be found in the review articles[12,13]. Engineering of twinning structure enables the development of functional materials with unprecedented properties. For example, the presence of dense nanoscale CTBs not only increases the mechanical strength significantly but also retains the high electrical conductivity of copper, because the CTBs are effective dislocation blockers but not strong electron scatters[14]. One recent study has shown that Kirkendall voiding is suppressed at the interface between a solder alloy and a <111>-oriented copper film with dense CTBs in parallel to film surface because excess vacancies are eliminated by abundant ITBs[15].

Nanoscale materials such as nanoparticles (NPs) and nanowires (NWs) usually have a limited capacity for lattice shift. When a grain boundary in a nanocrystalline Cu meets a surface, it forms a groove. When a pair of CTBs in a nanotwinned Cu meets a surface, they form a zig-zag type faceted structure. However, the latter and the former behave very differently in chemical reactions and voiding kinetics. It has been shown that the chemical reactivity of copper NWs (CuNWs) with dense CTBs intercepting the free surface varies with the spacing of CTBs[16]. The smaller the CTB spacing, the better the corrosion resistance.

In this paper, we report the oxidation of a CuNW as an example to demonstrate how cavitation in nanomaterials is eliminated by modifying the nanoscale surface structure with a high-density nanoscale CTBs. The CuNW surface turns into a faceted structure of a very low atomic step density when intercepted by abundant CTBs, which decreases the rates of copper oxidation and vacancy generation at copper/oxide interfaces. A slow and orderly vacancy absorption behavior at interfaces causes a lattice shift without void formation in the oxidation reaction.

## Results

**Oxidation-induced void formation in CuNWs.** CuNWs of 70–90 nm in diameter were electrodeposited in a porous anodic alumina oxide membrane by pulsed electroplating at low temperature. The CuNWs exhibits polycrystalline microstructure in the initial growth stage (<1 μm in length) and evolves into a feature of highly dense CTBs in the later stage (Supplementary Fig. 1). We have specifically prepared these two kinds of CuNWs

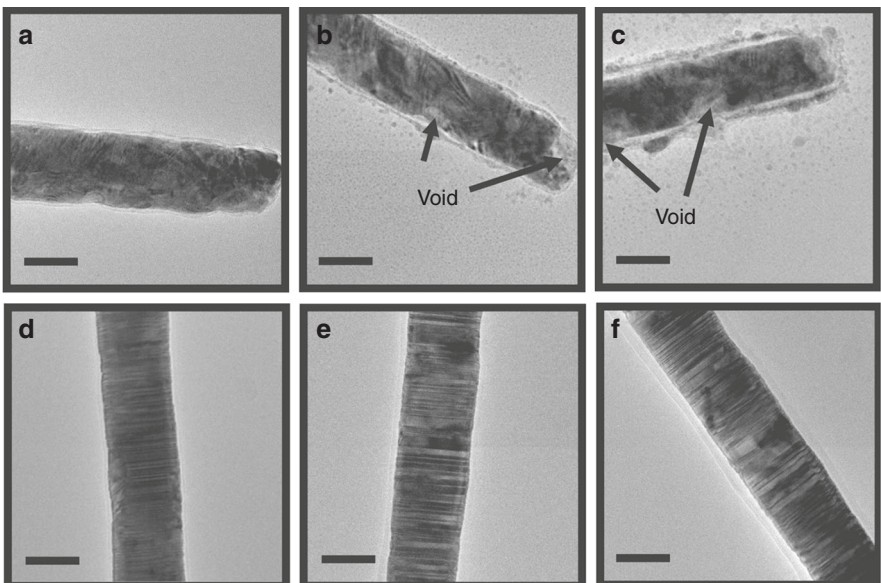

**Fig. 1** Bright-field TEM images of the copper nanowire (CuNWs). **a–c** Nanocrystalline CuNW after air-exposure for less than 1 day (**a**), 3 days (**b**) and 7 days (**c**). **d–f** Nanotwinned CuNW after air-exposure for less than 1 day (**d**), 3 days (**e**) and 7 days (**f**). Scale bar: 50 nm

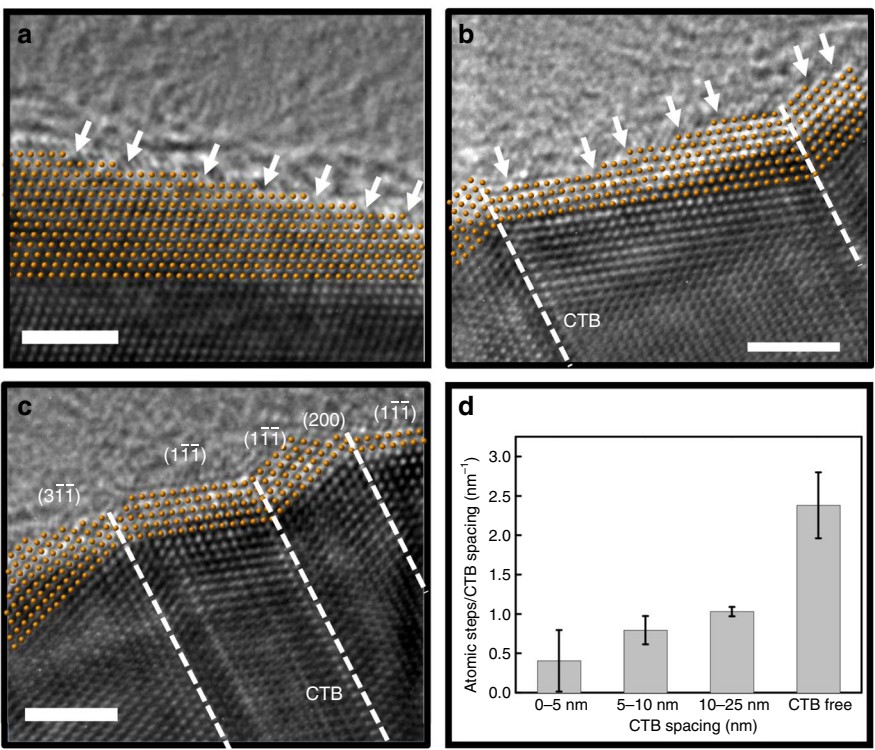

**Fig. 2** Atomic surface structure of fresh copper nanowires (CuNWs). **a–c** HRTEM images of the fresh CuNWs: twin-free region (**a**); twinned region with wide CTB spacing (**b**); twinned region with narrow CTB spacing (**c**). The white arrows indicate the atomic steps at the CuNW surface. Scale bar: 2 nm. **d** The statistics of atomic steps on the CuNW surface of various CTB spacings. The capped lines indicate the range of measurements recorded for each category

with different microstructures for direct comparison because both were prepared in the same run. One has dense CTBs perpendicular to the growth direction of the CuNW, while the other exhibits a nanocrystalline structure without nanotwin (Supplementary Fig. 2). If it is desirable, we can prepare each one of them in a separated CuNW.

The CuNWs were released from the membrane and dispersed onto a Mo-grid transmission electron microscopy (TEM) specimen holder. These CuNWs were examined by high-resolution TEM (HRTEM) after exposed to air ambient at room temperature for different durations. Figure 1 shows a set of bright-field TEM images taken from these two CuNWs after exposed to air ambient for 0, 3, and 7 days, respectively. Both the fresh CuNWs have a very thin native oxide layer (Fig. 1a, d). The oxide layer thickened slightly after air-exposure for 3 days. We observed that the nanocrystalline CuNW transformed gradually into a rough shell structure with some voids formed underneath the oxide layer (Fig. 1b, c). Surprisingly, while the nanotwinned CuNW is passivated by a uniform oxide layer, it retains its structure integrity (Fig. 1e, f), meaning it does not show void formation. Both the nanotwinned and nanocrystalline CuNWs were also heated at 150 °C in air ambient for 1 h and examined consequently by TEM (Supplementary Fig. 3). The oxide layer formed at elevated temperature was found to be slightly thicker than that formed at room temperature. Nevertheless, the nanotwinned CuNW still retained its structural integrity, while the nanocrystalline CuNW evolved into a hollow nanotube. The finding suggests that the nanotwinned structure is stable against oxidation and void formation even at elevated temperature.

Generally, the thin oxide layer formed in air ambient at low temperature is a $Cu_2O$ phase[17,18]. According to Cabrera and Mott hypothesis[19], a strong electric field that is established between

copper and absorbed oxygen will expedite the transport of Cu ions across the oxide layer, leading to the fast growth of cuprous oxide in the initial stage of oxidation. Because the outward diffusion of Cu ions is much faster than the inward diffusion of oxygen in $Cu_2O$[18], excess vacancies are generated on the copper side near the $Cu/Cu_2O$ interface. If the excess vacancies are not fully absorbed by lattice defects, such as dislocations and GBs, local vacancy super-saturation may trigger the nucleation of voids in the CuNW. After forming stable void nuclei, an ensuing outward flux of Cu ions will sustain the growth of voids in the CuNWs (Fig. 1b, c). It is worth noting that as $Cu_2O$ is not a protective oxide such as $Al_2O_3$ on Al, the interface between $Cu_2O$ and Cu is relatively effective as a source and sink of vacancies. So voids can form in nanocrystalline CuNWs. However, the random walk of vacancies during diffusion is blocked by the dense nanoscale CTBs. For this reason, the oxidation is retarded in nanotwinned CuNWs. Moreover, for very small NWs and NPs, a vacancy gradient exists between the inner and outer boundaries of the oxide shell due to the Gibbs-Thomson effect and it may also retard the inward diffusion of vacancies and prohibit void formation in the NWs and NPs; this is known as the inverse Kirekendall effect[20]. Because the nanotwinned and nanocrystalline CuNWs inspected have the similar diameter, the inverse Kirkendall effect should not be the major cause of lacking voids in the nanotwinned CuNW.

**Surface-structure controlled oxide growth**. Figure 2a–c shows respectively the HRTEM images taken from different regions of the fresh CuNWs. The twin-free region exhibits many atomic steps at the CuNW surface, as indicated by the white arrows (Fig. 2a). But, the twin-modified surface shows less atomic steps

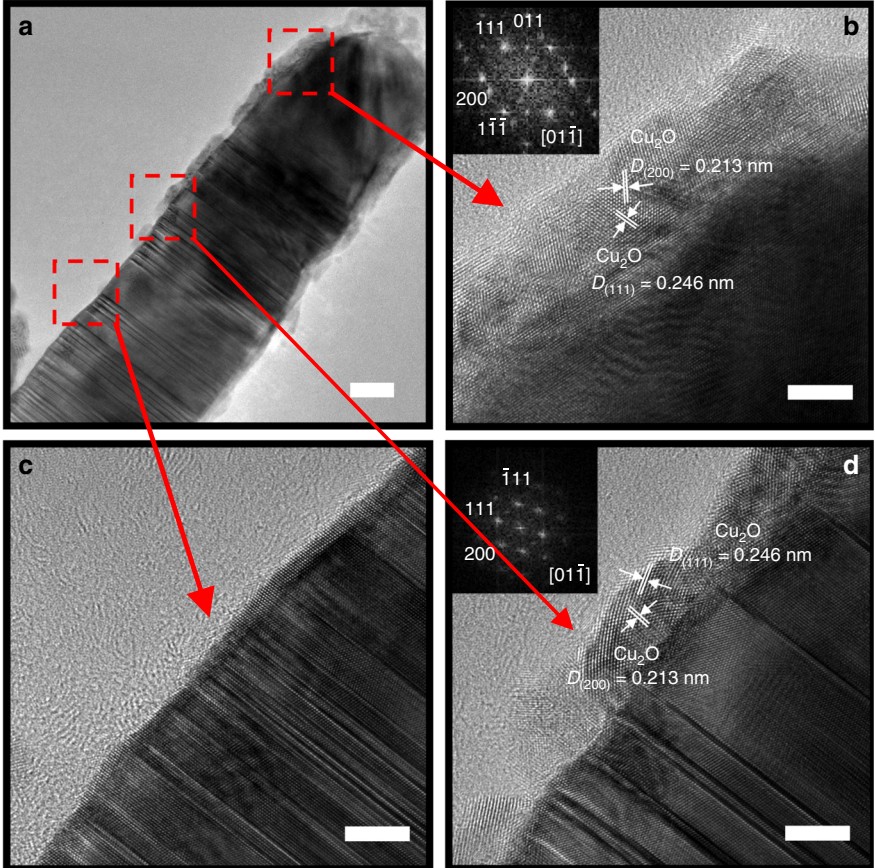

**Fig. 3** Morphology of crystalline $Cu_2O$ phase formed on a copper nanowire with different twinning features. **a** Low-magnification TEM image, scale bar: 20 nm; **b**–**d** HRTEM images of $Cu_2O$ phase in the region of large grain (**b**), narrow CTB spacing (**c**), and wide CTB spacing (**d**). Scale bar: 5 nm. The insets of **b** and **d** are electron diffraction patterns of $Cu_2O$ obtained by fast Fourier transform technique

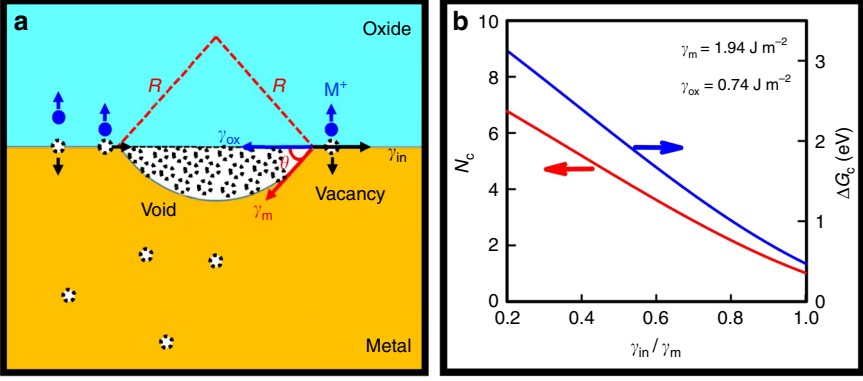

**Fig. 4** Vacancy clustering and absorption at the metal/oxide interface. **a** Vacancy clustering at flat metal/oxide interface; **b** Variation of the critical vacancy cluster size ($N_c$) and the energy barrier of void nucleation ($\Delta G_c$) with the ratio of interfacial energy ($\gamma_{in}$) and metal surface energy ($\gamma_m$)

and becomes faceted boundaries composed of low-index planes (Fig. 2b, c). To provide a quantitative comparison of atomic steps at the twin-modified surface, we counted the number of atomic steps at the surface and normalized the number by the CTB spacing. Figure 2d shows the normalized atomic steps of the twin-modified surface categorized by CTB spacing: 0–5; 5–10; 10–25 nm; CTB-free. The data from each category were obtained from 5 to 7 different locations of the HRTEM images taken from the CuNWs inspected. We note that the smaller the CTB spacing, the lesser normalized atomic steps at the twin-modified surface. The

surface with very narrow CTB spacing (<5 nm) sometimes exhibits an atomically flat surface (Fig. 2c). Previous studies of single-crystal surfaces of noble metals have indicated that the high-index crystal planes generally exhibit a high catalytic activity due to a high density of atomic steps and dangling bonds[21,22]. The CuNW surface with narrowly spaced CTBs exhibits a lower atomic step density, hence it is more chemically stable against oxidation.

The influence of atomic surface structure on Cu oxidation is also reflected by the growth of cuprous oxide on the nanotwinned

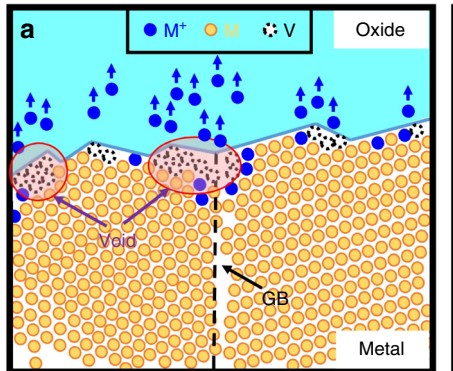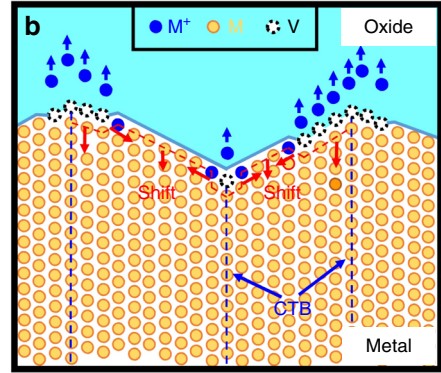

**Fig. 5** Schematics of vacancy transport and sinking operation associated with metal oxidation. **a** Vacancy clustering at polycrystalline metal/oxide interface; **b** vacancy absorption at nanotwinned metal/oxide interface

CuNW. Figure 3 shows the morphology of crystalline Cu₂O phase formed on a CuNW with different twinning features. At the right-hand side of the CuNW, the CTB-free region possesses the diversely oriented and thick cuprous oxide crystallites (Fig. 3b). Furthermore, the region of narrowly spaced CTBs exhibits a much thinner cuprous oxide than that with coarse twin lamellae (Fig. 3c, d). The trend of increasing oxide growth rate agrees with increasing atomic step density with increasing CTB-spacing for the twin-modified CuNW surfaces (Fig. 2d). In-situ TEM study has also provided a direct evidence that the nucleation and growth of cuprous oxide on a roughened Cu (110) surface is much faster than that on a flat Cu (110) surface[23]. The faceted surface between narrowly spaced CTBs possesses a slow oxidation rate, which may enable the epitaxial growth of cuprous oxide on Cu. If the growth of cuprous oxide is limited by the outward diffusion of copper cations in oxide, the epitaxial oxide layer would have a reduced growth rate due to the lack of GBs serving as fast diffusion paths. It explains why the twin-modified surface structure affects the growth kinetics of cuprous oxide on CuNWs.

**Vacancy nucleation and absorption at copper/oxide interfaces.** Next, we consider the voiding and no-voiding phenomena in oxidizing CuNWs. The Kirkendall effect consists of two competing phenomena: Kirkendall void formation and Kirkendall lattice shift[24]. The latter leads to no void formation and the displacement of the original interface (Kirkendall plane) relative to the Matano plane in a volume-fixed frame leads to marker motion. The former is driven by vacancy super-saturation and followed by void nucleation and growth. Consider the void nucleation at the metal/oxide interface, the vacancies generated as a result of the injection of metal cations ($M^+$) into the oxide may either diffuse into the underlying metal or accumulate at the metal/oxide interface (Fig. 4a). The critical vacancy cluster size ($N_c$) and the energy barrier of void nucleation ($\Delta G_c$) are given by.

$$N_c = \frac{\pi}{3\Omega_V}\left[\frac{2\gamma_m}{G_V^f/\Omega_V}\right]^3 \times \left[2 - 3\left(\frac{\gamma_{in} - \gamma_{ox}}{\gamma_m}\right) + \left(\frac{\gamma_{in} - \gamma_{ox}}{\gamma_m}\right)^3\right]$$

(1)

$$\Delta G_c = \frac{4\pi}{3}\left[\frac{\gamma_m^3}{(G_V^f/\Omega_V)^2}\right] \times \left[2 - 3\left(\frac{\gamma_{in} - \gamma_{ox}}{\gamma_m}\right) + \left(\frac{\gamma_{in} - \gamma_{ox}}{\gamma_m}\right)^3\right]$$

(2)

where $G_V^f$ is the vacancy formation energy, $\Omega_V$ is the atomic volume of metal, $\gamma_m$, $\gamma_{ox}$, and $\gamma_{in}$ are metal surface energy, oxide surface energy and metal/oxide interfacial energy, respectively.

Assuming $\gamma_{ox} = 0.74\,\mathrm{J\,m^{-2}}$ (in ref. [25]), $\gamma_m = 1.94\,\mathrm{J\,m^{-2}}$ and $G_V^f = 0.92\,\mathrm{eV}$ (in ref. [26]), we have calculated the values of $N_c$ and $\Delta G_c$ as a function of $\gamma_{in}/\gamma_m$ ratio for void nucleation at the copper/oxide interface (Fig. 4b). Both $N_c$ and $\Delta G_c$ are found to decrease with the increase of $\gamma_{in}/\gamma_m$ ratio. We note that the Cu/Cu₂O interfacial energy is the sum of misfit strain energy and chemical interaction energy of the interfacial region[27]. The lattice misfit between Cu and Cu₂O is around 15%, implying a large interfacial strain energy, even though the strained structure can be relaxed to some extent by forming misfit dislocations[28]. The chemical interaction energy usually depends on the composition of contacting materials, but it is also modulated by nanostructured interfaces[29]. Presumably, the interfacial energy would vary with the atomic structure at Cu/Cu₂O interface. Assuming $\gamma_{in}$ is about one-half of $\gamma_m$, the critical vacancy cluster size is ~4 and the nucleation energy barrier is 2 eV. It is worth mentioning that the above calculation does not take the curved surface and triple junction sites of CuNWs into consideration, which are expected to give an even lower nucleation energy[30]. Such a small nucleus size and low activation energy indicate that void nucleation should not be the rate-limiting process for void formation at the Cu/Cu₂O interface. Instead, vacancy generation, transport and absorption operation may play a more important role in void formation.

After forming a native surface oxide layer, the subsequent oxidation reaction will transform metal atoms (M) into metal cations ($M^+$) at the metal/oxide interface. The injection of metal cations into the oxide will leave a vacancy at the interface. The metal atoms at the step and kink sites, in general, are more susceptible to oxidation than those at a flat surface. Thus, the vacancy generation rate shall increase with the density of atomic steps at interface. Vacancy transport along the interface is fast and should not be the rate-limiting step. Consequently, vacancy absorption becomes essential for void formation. For a polycrystalline metal of high atomic step density and full of triple junction sites, a high vacancy generation rate and a low nucleation threshold enable the quick formation of voids at the interface (Fig. 5a). On the other hand, although the twin-modified interface has low vacancy generation rate, voids would still form in case of lacking effective vacancy sinks. Demkowicz et al.[31] have pointed out that CTBs are not effective vacancy sinks and do not curtail the formation of vacancy clusters based on their theoretical and experimental investigation of radiation-induced defect clusters in nanotwinned copper. The next possible vacancy sinks will be metal/oxide interface. The extremely low vacancy generation rate, accompanied by the effective vacancy absorption operation at the interface, facilitate the layer-by-layer removal of metal atoms of CuNWs (Fig. 5b). Therefore, instead of forming voids, a lattice plane is removed at the interface and the

displacement of lattice plane (Kirkendall shift) occurs accordingly. Our finding suggests that the twin-modified CuNW surface not only changes interfacial reaction rate but also alters the voiding kinetics at the Cu/Cu₂O interface.

## Discussion

In conclusion, we have demonstrated the suppression of Kirkendall voiding in the CuNW during oxidation reaction by modifying the surface with the introduction of dense CTBs. We found that the twin-modified surface exhibits a faceted structure composed of low-index planes and also the density of atomic steps decreases with decreasing CTB spacing. The faceted interface leads to slow rates of Cu ionization and vacancy formation at the Cu/Cu₂O interface. The suppressed vacancy formation rate together with the highly effective vacancy absorption at the interface lead to lattice plane shift without forming voids in the nanotwinned Cu NWs.

## Methods

**Materials**. Aluminum foils (Al, 99.997%) and Nickel chloride hexahydrate (NiCl$_2$·6H$_2$O, 99.3%) were purchased from Alfa Aesar. Perchloric acid (HClO$_4$, 70%), and Potassium dichromate (K$_2$Cr$_2$O$_7$, 99.5%) were purchased from Sigma-Aldrich. Oxalic acid dihydrate (C$_2$H$_2$O$_4$·2H$_2$O, 99.5%), Sodium hydroxide (NaOH, 97%) and copper sulfate (CuSO$_4$, 99.5%) were purchased from Showa Kako. Hydrochloric acid (HCl, 37%) was purchased from Scharlau. Phosphoric acid (H$_3$PO$_4$, 72%) was purchased from J.T. Baker® chemicals. Acetone (C$_3$H$_6$O, 99%), isopropyl alcohol (C$_3$H$_7$OH, 99%), and anhydrous ethanol (C$_2$H$_5$OH, 99.5%) were purchased from Echo Chemical. Epoxy quickstick was purchased from EMS. The deionized water was obtained from the EASYpure® II system. All the materials were used as received without further purification.

**Preparation of anodic aluminum oxide membrane**. A porous aluminum oxide (AAO) membrane was used as a temple for subsequent electrodeposition of CuNWs. Aluminum foils of 0.2 mm in thickness were cleaned and rinsed consectively with acetone, isopropyl alcohol, and deionized water. To achieve a smooth surface for the following anodization process, the Al foils were electropolished in a 4: 1 v/v mixture solution (300 ml) of anhydrous ethanol and perchloric acid at a voltage of 40 V for 20 s at 10 °C using an electrochemical analysis system (Jiehan 5000, Jiehan Tech.). The AAO membrane were prepared through a two-step anodization process. The first anodization was conducted at a voltage of 40 V in an oxalic acid solution (250 ml, 0.3 M) at 10 °C. A selective etching was performed to reveal a hexagonal pattern on the first AAO layer using a mixture solution (100 ml) of phosphoric acid (6 g) and potassium dichromate (1.8 g) at 60 °C for 30 min. A second anodization process same as the first one was carried out for 12 h to obtain a 40 μm-thick AAO layer. A phosphoric acid solution (250 ml, 5 wt.%) was used to remove the oxide layer at the bottom of nanopores in the AAO. Next, a 200 nm-thick Ni layer was vaporated at one side of AAO by e-gun deposition as a contact electrode for subsequent electroplating of CuNWs. The sample (Ni side) was glued on a glass substrate by epoxy on a hot plate at 135 °C. The exposed AAO and residual Al was sequentially removed by a sodium hydroxide solution (250 ml, 1 M) and a mixture of hydrochloric acid (100 ml) and NiCl$_2$ solution (100 ml, 0.5 M) at room temperature, respectively. The AAO membrane with open pores of 70–90 nm in size was then obtained.

**Electrodeposition of CuNWs**. The electrodeposition of CuNWs were carried out in a standard three-electrode cell by setting the sample at working electrode, a graphite bar at counter electrode and a saturated calomel electrode (SCE) at reference electrode, respectively. The pH value of copper sulfate plating solution (300 ml, 0.7 M) is around 3~4. The CuNWs were deposited into the AAO membrane at 0 °C under a pulse current of 0.4 A cm$^{-2}$ in peak density with a duty cycle of 0.02 s (on)/1 s (off). The height of CuNWs was around 30 μm after supplying 4000 cycles of pulse current.

**Characterization**. The CuNWs were released from the AAO membrane by immersing the sample in sodium hydroxide solution (250 ml, 1 M) for 50 min. The CuNWs were collected and kept in anhydrous ethanol. Samples for TEM were prepared by dropping the CuNWs suspension onto a Mo-grid specimen holder that was coated with a carbon film, and dried under ambient condition for 20 min. These CuNWs were examined by high-resolution TEM (HRTEM, JEM-3000F, JEOL) after exposed to air ambient at room temperature for different durations. Two different types of CuNWs were identified and located on the Mo-grid TEM specimen holder: one with dense CTBs perpendicular to the CuNW growth direction, and the other with a nanocrystalline feature (Supplementary Fig. 1, Supplementary Information). These CuNWs were examined by high-resolution

TEM (JEM-3000F, JEOL) operated at 300 kV. The HRTEM images were taken in the [011] direction of the CuNWs inspected.

**Data availability**. The data that support the findings of this study are available from the corresponding authors upon request.

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

## Acknowledgements

We acknowledge the support from Ministry of Science and Technology, Taiwan through the grant MOST 102-2221-E-007-048-MY3 and MOST 105-2221-E-007-016-MY2.

## Author contributions

C.-L.H. fabricated samples and performed TEM inspection and atomic surface step analysis. W.-L.W. contributed to the sample preparation and TEM analysis. K.N.T. analyzed the results. C.N.L. supervised and coordinated the work, and wrote the manuscript with input from all authors.

## Additional information

**Competing interests:** The authors declare no competing financial interests.

