## [Peer Review File · Nature Communications]

Editorial Note: Parts of this peer review file have been redacted as indicated to remove third-party material where no permission to publish could be obtained.

Reviewers' Comments:

Reviewer #1:

Remarks to the Author:

The authors have demonstrated a novel finding regarding chemical reactivity of nanotwinned structures in Cu nanowires. The faceted surfaces due to the closely spaced coherent twin boundaries intersecting the surface leads to slow Cu ionization and vacancy formation at the Cu/Cu₂O interface. The overall effect seems to be suppression of Kirkendall voids.

The authors should comment on the applicability of this mechanism at above ambient temperatures. How stable is the CTB structure and the surface topology generated by it at elevated temperatures, even in the 100 -200 °C range?

Also, will this mechanism apply to any nanotwinned metal or is there something special about the Cu/Cu₂O interface regarding the ability of the interface to absorb vacancies?

Given the broad research interest in nanotwinned metallic structures, it will be helpful to cite some recent reviews on this topical area (ANNUAL REVIEW OF MATERIALS RESEARCH, VOL 44 Pages: 329-363 (2014); and MRS Bulletin April 2016 issue).

Reviewer #2:

Remarks to the Author:

The main discovery of this article is that the introduction of dense coherent twin boundaries can effectively suppress interdiffusion-induced void formation (the Kirkendall Effect) during the oxidation of copper nanowires. This reviewer has a number of concerns about the significance and potential impact that may prevent recommendation of this article for publication in Nature Communications.

First, the potential impact seems to be limited. Suppression of voiding is a very important topic for bulk scale materials, but its significance for nanoscale structures is not fully demonstrated in this article. In the Introduction, the authors mention the requirement of stable and void-free structure in the fabrication of VLSI devices. The impact of this work will be much improved if the authors can build prototypes of VLSI devices using void-containing and void-free nanowires and compare their performances in connection to their void/defect structures.

Second, the potential impact of this paper is further limited by the fact that this work does not provide an effective method for introducing dense coherent twin boundaries to copper nanowires in a controllable manner. Nanowires with twin boundaries were produced together with those without, and then picked up based on random tries.

Third, this article is not the first one reporting the oxidation prevention behavior of metals with low-angle grain boundaries and low-energy coincident site lattice boundaries.

Reviewer #3:

Remarks to the Author:

Recommendation: Minor Revisions

This manuscript reports the suppression of Kirkendall voiding in oxidation of Cu nanowires by introducing nanotwins. Kirkendall voiding is a consequence of the different diffusivities of atom in a diffusion couple and has been utilized as an effective and popular approach to prepare hollow/porous structures. Suppression of Kirkendall voiding, on the other hand, is also technically important as voids are undesirable in some specific applications such as microelectronic devices. Here the authors discovered that the nanotwins in Cu nanowires could prohibit the voids formation because of the relatively flat surface and the low vacancy generation rate at the Cu/oxide

interface. The few atomic steps with less activity also slow the oxidation rate. The data presented are clear and the findings are interesting. The manuscript was well written. Overall, a good piece of work, suitable for publication in Nature Communications after the following minor issues have been addressed:

1. The suppression of Kirkendall void formation has also been observed in many solder reactions. For example, nanotwin boundaries in Cu has shown to prevent the Kirkendall void formation at the interfaces of Cu-Sn joint. Even though the subject studied here is in nanoscale size, the mechanisms could still be similar. The authors should further discuss and compare the roles of twin boundaries played in the elimination of voids in bulk materials and in nanomaterials.
2. The voids formation at interfaces sometimes is not solely caused by Kirkendall effect but could also due to the decomposition of impurities. The electroplating process could possibly introduce impurities (for example, S) into Cu nanowires. Have the authors examined the impurities on nanocrystalline surface and nanotwinned surface? These two types of surface may behavior differently in adsorption of impurities due to their different surface reactivity.
3. The authors claimed that the metal/oxide interface is possible vacancy sinks that adsorb vacancies. What is the difference between nanocrystalline metal/oxide and nanotwinned metal/oxide interfaces? Why the latter is a more effective vacancy adsorbent? These issues should be further clarified.

We highly appreciate the comments and suggestions provided by the reviewers, which improve the quality of the paper remarkably. The point-to-point response to the reviewers' comments are shown below.

Reviewer #1 Comments:

The authors have demonstrated a novel finding regarding chemical reactivity of nanotwinned structures in Cu nanowires. The faceted surfaces due to the closely spaced coherent twin boundaries intersecting the surface leads to slow Cu ionization and vacancy formation at the Cu/Cu₂O interface. The overall effect seems to be suppression of Kirkendall voids.

The authors should comment on the applicability of this mechanism at above ambient temperatures. How stable is the CTB structure and the surface topology generated by it at elevated temperatures, even in the 100 -200 °C range?

Response: Both the nanotwinned and nanocrystalline CuNWs were also heated at 150 °C in air ambient for 1 h and examined consequently by TEM (Fig. S3). The oxide layer formed at elevated temperature was found to be slightly thicker than that formed at room temperature. Nevertheless, the nanotwinned CuNW still remained its structure integrity, while the nanocrystalline one evolves into a hollow nanotube. The finding suggests that the nanotwinned structure indeed is stable against oxidation and void formation even at elevated temperature.

Fig. S3 Bright-field TEM images of the nanotwinned and nanocrystalline CuNWs. (A, B) before thermal treatment; (C, D) after heating at 150 °C in air for 1 h. The inset of (D) is the selective area electron diffraction pattern of the nanocrystalline CuNW with oxide shell.

Also, will this mechanism apply to any nanotwinned metal or is there something special about the Cu/Cu₂O interface regarding the ability of the interface to absorb vacancies?

Response: Whether voids occur in the CuNW depends on the vacancy generation and

annihilation at Cu/Cu₂O interface. The nanotwin-modified interface possesses a low atomic step density and hence low vacancy generation rate. As described in the paragraph right before the conclusion, it is the extremely low vacancy generation rate, accompanied by the effective vacancy absorption at the interface, facilitating the layer-by-layer removal of metal atoms of CuNWs (Fig. 5b). As long as the nanotwinned metals possess faceted surfaces (low atomic step density) with operative interfacial vacancy sinking mechanism, Kirkendall voiding should be suppressed during the oxidation reaction.

Given the broad research interest in nanotwinned metallic structures, it will be helpful to cite some recent reviews on this topical area (ANNUAL REVIEW OF MATERIALS RESEARCH, VOL 44 Pages: 329-363 (2014); and MRS Bulletin April 2016 issue).

Response: The text in in the introduction part has been modified below to address recent work on nanotwinned metallic materials with the references indicated.

“Recent experimental and modeling work towards understanding the effects of nanotwinned structure on mechanical and physical properties of metallic materials can be found in the review articles^{12,13}. Engineering of twinning structure enables the development of functional materials with unprecedented properties.” Two references are added.

Reviewer #2 Comments:

The main discovery of this article is that the introduction of dense coherent twin boundaries can effectively suppress interdiffusion-induced void formation (the Kirkendall Effect) during the oxidation of copper nanowires. This reviewer has a number of concerns about the significance and potential impact that may prevent recommendation of this article for publication in Nature Communications.

First, the potential impact seems to be limited. Suppression of voiding is a very important topic for bulk scale materials, but its significance for nanoscale structures is not fully demonstrated in this article. In the Introduction, the authors mention the requirement of stable and void-free structure in the fabrication of VLSI devices. The impact of this work will be much improved if the authors can build prototypes of VLSI devices using void-containing and void-free nanowires and compare their performances in connection to their void/defect structures.

Response: Voids are undesirable for many nanoelectronic devices and nanoelectromechanical systems. For example, copper interconnects in very-large-scale-integration devices require a stable and void-free structure for performance and reliability considerations. The presence of voids can cause circuit opening which is the worst case of device failure. Semiconductor industry has devoted tremendous efforts to prevent voiding in interconnects during processing and subsequent operation because the cross-section of copper interconnects is in nanoscale today. That’s why we believe that the prevention of Kirkendall void formation is of

keen interest in nanotechnology. We agree that the implementation of nanotwinned CuNWs in practical VLSI devices would be a good way to validate the performance of nanotwinned CuNWs. However, the building of prototypes of VLSI devices is beyond the capability of our campus lab by considering the processing tool availability and complicated device layout and integration. Nevertheless, we would seek for joint research effort from some semiconductor manufacturers to realize the goal in our future study.

Second, the potential impact of this paper is further limited by the fact that this work does not provide an effective method for introducing dense coherent twin boundaries to copper nanowires in a controllable manner. Nanowires with twin boundaries were produced together with those without, and then picked up based on random tries.

Response: The dense CTBs indeed can be introduced into the CuNWs in a controllable way. It has been reported that the microstructure, growth direction and CTB spacing in electroplated CuNWs can be controlled by electrodeposition parameters, e.g. pulsed current density and electrolyte temperature, as shown in the Fig. 4 of the reference (T. C. Chan et al, *Nanoscale*, 6, 7332 (2014)). In this study, the CuNW grown in the AAO template exhibits polycrystalline microstructure in the initial growth stage ($< 1 \mu\text{m}$ in length) and evolves into a feature of highly dense coherent twin boundaries in the later stage (Fig. S1 in the supplementary information.) The inspected nanocrystalline and nanotwinned structure were obtained from different regions of the CuNWs.

Third, this article is not the first one reporting the oxidation prevention behavior of metals

with low-angle grain boundaries and low-energy coincident site lattice boundaries.

Response: We checked the reference about the oxidation resistance of nanotwinned Cu (Y. Zhao et al, Acta Mater. 67 (2014) 181–188). They reported that in highly nanotwinned Cu, some of the columnar GBs have alternating segments of high-angle and low-angle GBs due to the alternating nanotwins, while some other columnar GBs are primarily high-angle GBs. It is the low-angle GB segments that suppress the intergranular corrosion and improve the oxidation resistance of nanotwinned Cu. However, there is no physical model provided to explain the presence of these two distinct configurations of GB network and how to modulate the appearance of low-angle GB segments. Although twin-modified nanowire surface may share the similar mechanism as twin-modified GBs, we provide more direct evidence showing how to modulate the surface structure by varying CTB spacing. In this report, the key point we try to make is that **the dense internal defect (dense nano-twin boundary) will tailor the surface structure and alter the chemical reaction and voiding kinetics in Cu/Cu₂O core-shell structure.** We believe that the aspect of internal defect modulated surface microstructure has not been addressed in other nanowires so far. We emphasize this point as described in the last paragraph of introduction part. – “When a grain boundary in a nanocrystalline Cu meets a surface, it forms a groove. When a pair of twin boundaries in a nanotwinned Cu meets a surface, they form a zig-zag type faceted structure. However, the latter and the former behave very differently in chemical reactions and voiding kinetics”, as addressed here but not by anyone before.

Reviewer #3 Comments:

Recommendation: Minor Revisions

This manuscript reports the suppression of Kirkendall voiding in oxidation of Cu nanowires by introducing nanotwins. Kirkendall voiding is a consequence of the different diffusivities of atom in a diffusion couple and has been utilized as an effective and popular approach to prepare hollow/porous structures. Suppression of Kirkendall voiding, on the other hand, is also technically important as voids are undesirable in some specific applications such as microelectronic devices. Here the authors discovered that the nanotwins in Cu nanowires could prohibit the voids formation because of the relatively flat surface and the low vacancy generation rate at the Cu/oxide interface. The few atomic steps with less activity also slow the oxidation rate. The data presented are clear and the findings are interesting. The manuscript was well written. Overall, a good piece of work, suitable for publication in Nature Communications after the following minor issues have been addressed:

1. The suppression of Kirkendall void formation has also been observed in many solder reactions. For example, nanotwin boundaries in Cu has shown to prevent the Kirkendall void formation at the interfaces of Cu-Sn joint. Even though the subject studied here is in nanoscale size, the mechanisms could still be similar. The authors should further discuss and

compare the roles of twin boundaries played in the elimination of voids in bulk materials and in nanomaterials.

Response: A reference regarding the suppression of Kirkendall void formation between a solder alloy and a <111>-oriented copper film with dense CTBs in parallel to film surface has been cited in the introduction [Ref. 15]. It has been suggested that the excess vacancies resulting from unequal Cu and Sn atomic fluxes during soldering reaction are mainly absorbed by abundant ITBs in the nanotwinned Cu film.

In regard to the Kirkendall effect in bulk materials and nanoscale materials, we have revised the introduction part of the manuscript as below.

“The major difference between Kirkendall effect in bulk materials and nanoscale materials is due to the effect of lattice shift. In bulk materials, lattice shift can lead to vacancy equilibrium everywhere in the sample. In nanoscale materials, lattice shift tends to be negligible and the vacant lattice sites condense to form voids. Voids are undesirable for many nanoelectronic devices and nanoelectromechanical systems. For example, copper interconnects in very-large-scale-integration devices require a stable and void-free structure for performance and reliability considerations. Therefore, how to prevent Kirkendall void formation is of interest in nanotechnology.”

2. The voids formation at interfaces sometimes is not solely caused by Kirkendall effect but could also due to the decomposition of impurities. The electroplating process could possibly introduce impurities (for example, S) into Cu nanowires. Have the authors examined the impurities on nanocrystalline surface and nanotwinned surface? These two types of surface may behave differently in adsorption of impurities due to their different surface reactivity.

Response: The CuNWs were prepared using copper sulfate solution without adding any surfactant or other additives. We have performed elemental analysis on nanocrystalline and nanotwinned CuNWs by TEM-EDS. The CuNWs were placed on Ni-grid TEM specimen holder to avoid the overlapping of EDS signals between Mo and S elements. The amount of S impurity is negligible with the signal level similar to the background noise according to the EDS spectrum and elemental mapping results shown below. No marked difference in S content between the nanocrystalline and nanotwinned CuNWs was observed. Therefore, the possible impact of adsorbed impurity, at least S element, on the oxidation and voiding kinetics of CuNWs can be excluded.

Elemental analysis of (A) nanocrystalline and (B) nanotwinned CuNWs by TEM-EDS.

3. The authors claimed that the metal/oxide interface is possible vacancy sinks that adsorb vacancies. What is the difference between nanocrystalline metal/oxide and nanotwinned metal/oxide interfaces? Why the latter is a more effective vacancy adsorbent? These issues should be further clarified.

Response: We have addressed this point by adding a short paragraph in the revision. (in the third paragraph of **Results and discussion**)

“It is worth noting that Cu_2O is not a protective oxide such as Al_2O_3 on Al, the interface between Cu_2O and Cu is relatively effective as source and sink of vacancies. So void can form in nanocrystalline CuNWs. However, the random walk of vacancies in diffusion is blocked by the dense nanoscale CTBs. For this reason, the oxidation is retarded in the part of nanotwinned CuNWs.”

Reviewers' Comments:

Reviewer #1:

Remarks to the Author:

The revised version is now acceptable for publication.

Reviewer #2:

Remarks to the Author:

The authors have partially addressed my concerns. However, my major concern regarding the impact of this work still remains. There is no question that void suppression is important to electronic devices in both bulk and nanoscale. However, it is not clear if the current study can provide an effective solution to the voiding problem at nanoscale devices. The authors did not provide any evidence to support that these nanotwinned nanowires could be implemented in practical nanoelectronic devices. A question along the line is if the nanotwins could remain stable under the working condition so that the void suppression could last for a long time. If the major significance of void suppression is in nanoelectronic devices and nanoelectromechanical systems, then some relevant data are needed.

Regarding the novelty, the authors argue that the current work provides more direct evidence than the previous report in how to modulate the surface structure by varying CTB spacing. I agree with the authors that the current work provides much-improved understanding, but the novelty is lower than expected for a paper in Nature Communication if the major point has been made previously.

Reviewer #3:

Remarks to the Author:

The authors have addressed my concerns. This paper is acceptable for publication.

We highly appreciate the comments and suggestions provided by the reviewers, which improve the quality of the paper remarkably. The point-to-point response to the reviewers' comments are shown below.

Reviewer #1 Comments:

The revised version is now acceptable for publication.

Response: Thanks to the reviewer for constructive discussion and valuable suggestions.

Reviewer #2 Comments:

The authors have partially addressed my concerns. However, my major concern regarding the impact of this work still remains. There is no question that void suppression is important to electronic devices in both bulk and nanoscale. However, it is not clear if the current study can provide an effective solution to the voiding problem at nanoscale devices. The authors did not provide any evidence to support that these nanotwinned nanowires could be implemented in practical nanoelectronic devices. A question along the line is if the nanotwins could remain stable under the working condition so that the void suppression could last for a long time. If the major significance of void suppression is in nanoelectronic devices and nanoelectromechanical systems, then some relevant data are needed.

Regarding the novelty, the authors argue that the current work provides more direct evidence than the previous report in how to modulate the surface structure by varying CTB spacing. I agree with the authors that the current work provides much-improved understanding, but the novelty is lower than expected for a paper in Nature Communication if the major point has been made previously.

Response: The reviewer's main concerns on this paper are the impact and the novelty of this work. On practical applications, there is not yet application in using nanotwinned Cu in interconnect technology on Si chips, but there is application of nanotwinned Cu in "redistribution layer" (RDL) in microelectronic packaging technology to improve its reliability because of no void formation. However, we cannot say much about it because it is related to industrial manufacturing. On novelty, while the reviewer admitted that the current work has provided improved understanding, but the novelty is lower than expected for Nature Communications. This comment is subjective. We do believe that the idea of changing the internal microstructure to affect the surface structure and surface property is very innovative. Our study successfully demonstrates how this idea works on the prevention of void formation in copper nanowires during the oxidation reaction. We should let the Editor to make a better

judgement.

Reviewer #3 Comments:

The authors have addressed my concerns. This paper is acceptable for publication.

Response: Thanks to the reviewer for constructive discussion and valuable suggestions.